# Fungal Infections of Implantation: More Than Five Years of Cases of Subcutaneous Fungal Infections Seen at the UK Mycology Reference Laboratory

**DOI:** 10.3390/jof8040343

**Published:** 2022-03-25

**Authors:** Andrew M. Borman, Mark Fraser, Zoe Patterson, Christopher J. Linton, Michael Palmer, Elizabeth M. Johnson

**Affiliations:** 1UK Health Security Agency National Mycology Reference Laboratory, Southmead Hospital, Bristol BS10 5NB, UK; mark.fraser@nbt.nhs.uk (M.F.); zoe.patterson@nbt.nhs.uk (Z.P.); chris.linton@nbt.nhs.uk (C.J.L.); michael.palmer@nbt.nhs.uk (M.P.); elizabeth.johnson@nbt.nhs.uk (E.M.J.); 2Medical Research Council Centre for Medical Mycology, University of Exeter, Exeter EX4 4QD, UK

**Keywords:** subcutaneous fungal infections, antifungal susceptibility, *Alternaria*, eumycetoma, phaeohyphomycosis, solid organ transplant

## Abstract

Subcutaneous fungal infections, which typically result from traumatic introduction (implantation) of fungal elements into the skin or underlying tissues, can present as a range of different clinical entities including phaeohyphomycosis, chromoblastomycosis, subcutaneous nodules or masses, and genuine eumycetoma. Here, we mined our laboratory information management system for such infections in humans and domestic animals for the period 2016–2022, including (i) fungal isolates referred for identification and/or susceptibility testing; (ii) infections diagnosed at our laboratory using panfungal PCR approaches on infected tissue; and (iii) organisms cultured in our laboratory from biopsies. In total, 106 cases were retrieved, involving 39 fungal species comprising 26 distinct genera. Subcutaneous infections with *Alternaria* species were the most frequent (36 cases), which possibly reflects the ubiquitous nature of this common plant pathogen. A substantial proportion of *Alternaria* spp. isolates exhibited reduced in vitro susceptibility to voriconazole. Notably, a significant number of subcutaneous infections were diagnosed in renal and other solid organ transplant recipients post transplantation, suggesting that humans may harbour “inert” subcutaneous fungal elements from historical minor injuries that present as clinical infections upon later immunosuppression. The current study underscores the diversity of fungi that can cause subcutaneous infections. While most organisms catalogued here were responsible for occasional infections, several genera (*Alternaria, Exophiala, Phaeoacremonuim, Scedosporium*) were more frequently recovered in our searches, suggesting that they possess virulence factors that facilitate subcutaneous infections and/or inhabit natural niches that make them more likely to be traumatically inoculated.

## 1. Introduction

Subcutaneous fungal infections are thought to result from traumatic implantation of the causative fungal organism into the subcutaneous tissue. As such, human cases of infection are observed more commonly in warmer climates and have been reported mainly in immunocompromised hosts [1,2,3,4]. Due to their foraging/hunting tendencies, similar infections are also frequently encountered in otherwise healthy domestic companion animals, in particular cats and dogs [5,6]. In humans, the exact disease presentation depends on the etiological agent involved but is also influenced to some extent by the exact host immunological status [7]. Subcutaneous phaeohyphomycosis is a localized infection caused by a wide and heterogenous range of dematiaceous (melanised) fungi and commonly presents as a well-encapsulated solitary nodule or subcutaneous mass on the extremities, near the site of previous trauma [8,9]. 

Chromoblastomycosis shares many similar features to phaeohyphomycosis: the etiological agents are melanised fungi and a number of distinct genera have been repeatedly implicated as causative agents and prevalence is, again, the greatest in tropical and sub-tropical locations. However, chromoblastomycosis can be distinguished from phaeohyphomycosis by the presence of distinctive muriform cells (sclerotic bodies) scattered singly or in clusters throughout granulomatous tissue, as reviewed in [10]. Moreover, while successful treatment of the localized nodules or subcutaneous masses typical of phaeohyphomycosis can often be achieved via surgical excision (with or without adjunctive antifungal therapy), the lesions associated with chromoblastomycosis are often more extensive and recalcitrant to treatment, with frequent recurrence [10]. Finally, eumycetoma is a chronic but progressive infection of the skin, subcutaneous tissues and eventually bone, characterised by the diagnostic triad of relatively painless but relentlessly enlarging subcutaneous mass, development of sinus tracts that exude sero-purulent discharge and the presence of fungal grains in discharge and tissue [11,12,13]. A growing number of distinct fungal agents of eumycetoma have been described to date, spanning several fungal orders [12,13]. 

All of the above manifestations present diagnostic and therapeutic challenges: the subcutaneous lesions/swellings are indolent in onset with generally slow progression, such that records of historical trauma at the site are often vague; medical intervention is also frequently sought relatively late in disease progression, especially in regions where access to healthcare can be challenging; and many of the agents associated with subcutaneous fungal infections exhibit reduced antifungal drug susceptibility in vitro and successful treatment usually requires surgical interventions coupled with protracted treatment with antifungal agents. The fact that many case reports and case series highlight the occurrence of all forms of subcutaneous mycoses in solid organ transplant recipients, and in particular renal transplant patients [7,14,15,16,17,18], further complicates treatment due to the challenges of employing many of the newer triazole antifungal agents in patients receiving tacrolimus.

The UK National Mycology Reference Laboratory (MRL), part of the UK Health Security Agency (UKHSA), provides a comprehensive service for the diagnosis and management of fungal disease. The MRL portfolio includes serological and fungal biomarker tests to aid the diagnosis of both superficial and invasive/systemic fungal infections, microscopy and culture and panfungal PCR analyses of respiratory secretions, biopsies and tissues where appropriate, identification and susceptibility testing of isolates of pathogenic yeast and moulds (filamentous fungi), both isolated at the MRL and referred from other laboratories, and therapeutic drug monitoring of serum drug concentrations in patients receiving antifungal therapy. Samples for testing are referred to the MRL from hospitals, microbiology laboratories and veterinary and general practitioner services across the UK. Here, we have analysed >5 years of MRL laboratory data to identify probable and proven cases of subcutaneous infections in humans and domestic companion animals, collating supporting clinical data, the methods of diagnosis/identification and antifungal susceptibility profiles of the causative agents (where available). As many cases of fungal keratitis also result from traumatic injury to the cornea [19,20], such cases were also retrieved during database analyses and included in the study. The present results underscore the wide range of causative agents and clinical presentations associated with subcutaneous fungal infections.

## 2. Materials and Methods

In order to compile a comprehensive list of all fungal infections of implantation referred to the MRL, the laboratory information management system (LIMS) was interrogated for the period October 2016 (the date of implementation of a new LIMS) through to 2 February 2022. Three different searches were performed in parallel: (i) all isolates of moulds (filamentous fungi) from all clinical human and animal specimens that had been submitted to the MRL for identification and susceptibility testing; (ii) all isolates of moulds that had been recovered at the MRL from human and animal clinical specimens processed at the MRL; and (iii) all panfungal PCR tests that had been performed at the MRL on human and animal samples where a positive PCR reaction was reported and successful identification of an organism was achieved by sequencing of PCR amplicons. These initial searches recovered 19,047 referred mould isolates, an additional 4919 mould isolates isolated at the MRL from the primary processing of clinical samples and 1854 positive panfungal PCR results. 

Each individual dataset was then hand-edited to retain only listings that corresponded to possible subcutaneous infections (including keratitis as this is often an infection of implantation/trauma) by excluding all listings corresponding to non-cutaneous, subcutaneous or ocular sites. The resulting edited datasets were then examined entry-by-entry for relevant accompanying clinical information that would support a genuine diagnosis of subcutaneous/implantation mycosis, resulting in final datasets containing 92 referred isolates of mould, 30 additional isolates recovered in culture at the MRL, and 59 infections diagnosed by panfungal PCR and sequencing. After compilation of the three datasets and sorting by patient/animal identifiers, 106 individual cases of subcutaneous fungal infection were retained (certain cases had repeat isolation of the same organism on multiple occasions, others were confirmed both by isolation in primary culture and panfungal PCR of biopsy/tissues). Finally, where mould isolates had been recovered or referred to the MRL and susceptibility testing requested, the LIMS was interrogated with individual MRL sample reference numbers and available antifungal susceptibility results and methods employed for the identification of isolates or additional diagnostic modalities were collected. 

All antifungal susceptibility testing at the MRL during this period was performed using the CLSI broth microdilution method M38-A2 [21] exactly as described previously [22], identification of isolates was performed by phenotypic examination, usually in combination with MALDI-TOF MS analyses and/or rDNA sequencing as described previously in [13,23]. The tissue processing, DNA extraction and panfungal PCR approaches employed during this period have also been described previously [24]; panfungal PCR was performed using the primers described previously, which target the D1/D2 regions of the 28S large ribosomal subunit, the ITS1 region and additional loci (actin, RNA polymerase second largest subunit and translation elongation factor 1α) where necessary [13,23].

## 3. Results

Searches of the MRL LIMS database for the period October 2016 through to 2 February 2022 returned a total of 106 separate cases of subcutaneous fungal infection or keratitis of likely traumatic origin. Cases involved 39 different fungal species encompassing 26 different genera (Table 1). Of note, 14/106 cases were reported in the recipients of solid organ transplants, most frequently renal transplant patients. For 34 of the 39 species implicated, searches retrieved 3 or less human or animal infections over the study period. However, *Alternaria* spp., *Scedosporium apiospermum, Exophiala* spp., *Madurella* spp. and *Medicopsis romeroi* were associated with significantly higher numbers of cases (42, 13, 9, 5 and 6 cases, respectively). This is perhaps unsurprising since *Alternaria* spp. are ubiquitous plant pathogens that are likely to be frequently associated with implantation injuries and keratitis in both humans and domestic animals who have accidents or aggressive interactions involving plant material, and *Scedosporium apiospermum* (a soil organism), *Exophiala* spp. and *Medicopsis romeroi* have frequently been associated in the existing literature with subcutaneous nodules and eumycetoma [5,6,7,8,9,13,16,17,18,20,23,25,26,27,28].

Table 2 describes the 64 non-*Alternaria* infections in more detail. All 38 species listed have been reported at least once previously as the causative agents of subcutaneous fungal infections or keratitis in humans or domestic animals. Indeed, at least 24 of the listed species have been consistently reported previously as significant causes of subcutaneous fungal infections, and in particular phaeohyphomycosis or eumycetoma. Moreover, 26 or the 64 cases in Table 2 (shown in bold) are definitively proven infections as evidenced by repeat isolations of the same organism from separate samples from usually sterile sites over many weeks, months or years, recovery of the same organism in culture and by panfungal PCR of separate tissue samples, positive histological appearance consistent with the organism identified, positive fungal biomarker testing, history of previous penetrating injury at the site in a region endemic for that particular organism, or a combination of the above. Of note, the majority of infections were diagnosed in male patients (61%; 39/64), and of the human cases 69% occurred in patients aged 50 years or over (Table 2).

Since successful treatment of cases of subcutaneous mycoses most frequently relies upon a combination of surgical intervention and often protracted antifungal therapy, antifungal susceptibility testing of the causative agent (if isolated in culture) is an important component of antifungal management in individual cases. In addition, the generation of antifungal susceptibility profile databases permits predictions of susceptibility/resistance in future cases involving the same species that are refractory to culture. Table 3 presents the antifungal susceptibility profiles of the non-*Alternaria* spp. isolated from subcutaneous infections during the period 2016–2022. The range of antifungal agents tested with each isolate was largely determined by the requesting physician and was further tailored to be appropriate for the type/site of infection. For example, isolates from cases of ocular infection (keratitis) were tested against additional topical antifungal agents that would be appropriate for the treatment of such presentations (natamycin, econazole). 

As there are no validated breakpoint interpretations for any of the antifungal agents for any of the isolates documented, we have loosely applied those epidemiological cut-off values and clinical breakpoints established for *Aspergillus fumigatus*, when available. These are: amphotericin B, itraconazole, voriconazole and isavuconazole ≤1.0 mg/L susceptible and >2.0 mg/L resistant, posaconazole ≤0.125 mg/L susceptible and >0.25 mg/L resistant. There are no validated anidulafungin breakpoints for moulds and the MICs against most of the moulds tested, with some notable exceptions, appear to be quite elevated. Terbinafine is usually used to treat dermatophyte infections and does not have validated breakpoints; for the treatment of moulds other than dermatophytes, for example in the treatment of highly refractory *Lomentospora prolificans* infections, it is usually only used in combination with voriconazole with which it demonstrates synergistic activity. As can be seen from the antifungal susceptibility data presented in Table 3, with the exception of *Fusarium solani* and some isolates of *Medicopsis romeroi*, when applying these breakpoints most organisms were susceptible to at least one of the triazole antifungals (itraconazole, posaconazole, voriconazole, isavuconazole) in vitro. However, no single triazole antifungal agent exhibited activity across the whole spectrum of organisms isolated from subcutaneous infections, underscoring the importance of accurate identification and susceptibility testing of individual isolates. 

By far the most common agent of subcutaneous infections retrieved in our LIMS searches were organisms in the genus *Alternaria*. A total of 42 cases were retained after consolidation of the three separate searches performed, corresponding to subcutaneous masses in domestic companion animals (*n* = 26), subcutaneous nodules in humans (*n* = 11), 6 of which were organ transplant recipients, and an additional 5 cases of fungal keratitis in humans (Table 1). Full details of clinical presentation and method of diagnosis are presented in Table 4. 

When agents are used topically such as in the treatment of mycotic keratitis, different higher breakpoints may be more applicable, as immediately after topical application concentrations of the antifungal agent should be greatly in excess of the MIC of the drug for the infecting organism. The three agents we tested that can be used topically for this indication are amphotericin B, available as a 0.15% solution (1500 mg/L) and voriconazole and natamycin, available as 1% solutions (10,000 mg/L). However, the duration that this localized concentration is maintained and penetration of the drug into tissue will impact on activity, so the validation of breakpoints in this setting has not been addressed.

As seen with infections due to the other agents of subcutaneous mycosis discussed above, there was an obvious sex bias in the human cases of traumatic *Alternaria* infection (9 infections in males, 5 in females [1 unknown sex]; 64% males), and again the majority of infections (13/16; 81%) were diagnosed in individuals >50 years of age. A similar sex bias was also observed in *Alternaria* spp. infections in animals (19 male: 6 female; 1 unknown; 76% male). Six of the 16 subcutaneous human infections occurred in post-solid organ transplant patients (four post-renal transplant, one post-hand transplantation, one organ not specified; Table 4), and were known to affect the extremities (predominantly the legs), in keeping with the usual sites of traumatic injury. Similarly, where stated, all *Alternaria* infections in domestic cats and dogs involved body sites likely to be prone to inoculation injuries (nose, ears, paws and legs), and in many cases (15/26) were confirmed by direct visualization of fungal elements in biopsy samples or histology, repeat isolation from independent samples, PCR-driven diagnosis or a combination of the above. 

The antifungal susceptibility profiles for the 12 *Alternaria* sp. isolates that were recovered in culture (cases 1–12 in Table 4) are shown in Table 5. The apparent relatively low rate of recovery of *Alternaria* spp. isolates (12 isolates from 42 cases) is a reflection of the sample types that were received in many cases, where clinical material was submitted after previous formalin fixation, or as wax curls from histological blocks, thus preventing attempts at culture. In most cases, examination of material by direct microscopy at the MRL, or histology reports that accompanied the samples revealed or reported fungal elements consistent with the final diagnosis, making it unlikely that these molecular diagnoses were detecting fungal contaminants rather than the true pathogen (data not shown). Based on epidemiological cut-off values and clinical breakpoints established for *Aspergillus fumigatus*, all isolates tested were susceptible to amphotericin B, itraconazole and anidulafungin (although only three isolates were tested with anidulafungin), and the majority had relatively low MIC values with posaconazole. However, voriconazole activity was severely reduced against all except two of the isolates, with MIC values that would be interpreted as intermediate (2 and 4 mg/L) or resistant (8 mg/L or above) based on *A. fumigatus* breakpoints. Similarly, although only three isolates were tested against isavuconazole, all three exhibited elevated MICs that would be above the usual range seen with *A. fumigatus*. This reduced in vitro susceptibility to voriconazole is not unique to *Alternaria* sp. isolates from subcutaneous infections, based on historically collated MRL antifungal susceptibility data, which demonstrated that 25/33 (76%) isolates from environmental or other sources would be classed as intermediate/resistant to voriconazole in vitro (data not shown). 

## 4. Discussion

Here, we have retrospectively searched the MRL LIMS to retrieve all possible cases of subcutaneous fungal infections and traumatic keratitis diagnosed at the laboratory between October 2016 and February 2022. Of the 105 cases recovered, 42 concerned *Alternaria* spp. (Table 1) with over 60% (*n* = 26) of those presenting as subcutaneous masses of the extremities in domestic companion animals, and a further 11 similar cases in humans (plus 5 cases of fungal keratitis). The preponderance of members of this genus as etiological agents of such infections probably reflects both its ubiquitous nature, and the fact that it is a common plant pathogen likely present on many hard or thorny materials on which humans and pets might injure themselves [29,30]. Indeed, subcutaneous infections with *Alternaria* species in humans and domestic animals have been reported extensively in the literature previously; in humans there is a clear association with previous solid organ transplantation [14,18,25,31,32,33,34]. This same pattern is reflected in the cases retrieved during this study: 6 of 11 human infections were in patients who had previously received solid organ transplants (Table 1 and Table 4). A substantial proportion of the *Alternaria* sp. isolates reported here displayed elevated in vitro MICs with voriconazole that would be indicative of resistance (5/11 isolates with MICs of 8 mg/L or higher: Table 5). Our own anecdotal evidence suggests that these elevated MICs with *Alternaria* and voriconazole might have clinical relevance, since we have been involved in the management of several cases of subcutaneous alternariosis that have failed to respond or progressed despite protracted voriconazole therapy and persistently therapeutic antifungal drug levels (AMB and EMJ, unpublished observations). Indeed, several published case reports of breakthrough/refractory *Alternaria* infections that occurred during voriconazole therapy would support this contention [33,35,36,37], as would individual studies reporting limited in vitro voriconazole activity against members of this genus [34,38]. However, cases of successful treatment of primary cutaneous *Alternaria* infections with voriconazole have also been reported [39], highlighting the importance of antifungal susceptibility testing of individual isolates to optimize therapeutic strategies. Additionally, it is possible that differences in MICs reported here and elsewhere [34,38] between individual *Alternaria* isolates could reflect species-specific differences in susceptibility. Effectively, *Alternaria* is a large and pleomorphic genus that comprises approximately 300 species separated into at least 25 taxonomic sections [40,41,42,43] with polyphasic approaches, including multi-locus sequence typing required for accurate identification to species level. Most published cases of human infections (including the present study) did not attempt such precise identification. Indeed, where molecular approaches were used to confirm identification of the isolates presented here, analyses were only sufficient to suggest that the majority of isolates likely belonged to *Alternaria* Section *Infectoriae* [42].

The 64 cases of infections of implantation that did not implicate *Alternaria* spp. involved an additional 38 species comprising 25 genera of ascomycetes, all of which have previously been reported as etiological agents of subcutaneous fungal infections or traumatic keratitis in humans (Table 1). The genera *Exophiala* (9 cases), *Madurella* (5 cases), *Medicopsis* (6 cases) and *Scedosporium* (16 cases), all of which have been frequently associated with subcutaneous fungal infection or genuine eumycetoma [5,6,7,8,9,13,16,17,18,20,23,25,26,27,28], predominated. Antifungal susceptibility profiles for the causative organisms again revealed variable susceptibility to amphotericin B and the most commonly employed systemic azole antifungal drugs (Table 3), again highlighting the importance of correct isolate identification and susceptibility testing of individual isolates in optimising patient management. Oral azole therapy is the preferred option in these patients wherever possible, as prolonged courses are often required. Together with the 6 cases of subcutaneous *Alternaria* spp. infection in organ transplant recipients, subcutaneous infections with other organism were identified in a further 8 solid organ transplant patients (7 renal transplant; 1 liver transplant). The responsible organisms were again varied and included *Madurella mycetomatis* (1), *Medicopsis romeroi* (2), *Parathyridaria percutanea* (2), *Phaeoacremonium rubrigenum* (2) and *Phialemoniopsis curvata* (1). According to the accompanying clinical details (where available), none of the 14 cases of subcutaneous infections in organ transplant recipients had evidence of previous infection at the affected body sites, nor were these cases of relapses of previously treated infections post-transplantation. Moreover, none of the patients had signs or symptoms of likely disseminated infection from a different primary site. These observations have several intriguing implications. First, several of the organisms reported here from subcutaneous infections in transplant patients are known agents of eumycetoma [12,13], and yet the clinical features of infections were more typical of phaeohyphomycosis or locally invasive fungal infection rather than genuine eumycetoma, due to the lack of draining sinuses, extensive tumefaction or production of fungal grains (see cases 3, 4 and 5 involving *Medicopsis romeroi*, Table 2). This suggests that the typical clinical presentation associated with subcutaneous fungal infections is likely to be determined also by the immunological status of the host rather than only by the etiological agent involved. Second, they imply that many of these infections diagnosed several years post-transplantation have arisen from subcutaneous fungal elements introduced during historical minor injuries that have remained latent and subclinical for many years until later immunosuppression. This concept is not entirely novel as several studies have reported re-activation of dimorphic fungal infection many years after initial exposure [43,44,45], immunological and epidemiological evidence has been presented in support of dormant cryptococcal infection [46] and anecdotal reports have described recurrence of previously treated infections following transplantation [47]. Since survival inside macrophages or formation of granuloma have been proposed to be prerequisites for fungal latency, it is perhaps not surprising that ubiquitous saprobes accidentally implanted into the sub-dermis (with subsequent granulomatous reactions) may also have the capacity to persist in inert form for many years in immunocompetent hosts. In this regard the dematiaceous nature of many of the agents identified is significant as the production of melanin in the cell wall has long been recognized as a putative virulence factor enhancing the survival of fungal cells within the host phagocytes [48,49,50]. While the bias towards male sex in subcutaneous fungal infections has previously been proposed to result from increased likelihood of engaging in physical outdoor work, the bias towards older age as reported here would also be in keeping with the idea that such infections may remain latent or clinically innocuous for many years. 

There are several limitations to the current study. It is unfortunate that for many of the cases described here, detailed history of previous trauma, occupation of the patients and geographical area of likely acquisition are lacking. In addition, for many of the solid organ transplantation patients, data concerning delay between transplantation and onset of clinical presentation and the nature and duration of immunosuppressive agents employed was lacking. In part, this reflects the nature of national reference laboratory work, which is based on patient/case referrals with limited direct access to patient data or the ability to seek additional clinical information. However, it is also a likely reflection of the fact that many subcutaneous infections present clinically many years after initial acquisition, following relatively minor or innocuous traumas that the patient does not recollect. This situation is probably aggravated in those cases that follow later solid organ transplantation, where an inoculated organism has remained inert/subclinical for many years. It is also unfortunate that for most of the cases, we cannot be certain that organisms submitted to antifungal susceptibility testing were from patients that were antifungal treatment naïve. However, given that these isolates were either referred to our laboratory from diagnostic biopsy specimens, or cultured from them here at the MRL, it is likely that the majority of these cases represent the initial presentation/diagnosis and that the patients had not received prior antifungal treatment (with the exception of the relapsed cases of *E. grisea* and *M. romeroi* (case 1) infection (Table 2). 

In summary, here we have presented over 5 years of data concerning fungal infections of implantation (subcutaneous fungal infections and traumatic fungal keratitis) referred to the UK MRL, together with antifungal susceptibility profiles of the causative agents, where available. While a wide array of fungal species and genera were implicated in such infections, a select few (*Alternaria* spp., *Exophiala* spp., *Madurella* spp., *Medicopsis romeroi* and *Scedosporium* spp.) predominated. Further studies will be required to explain the preponderance of these select organisms in subcutaneous infections in general and those affecting solid organ transplant recipients in particular. It is possible that this simply reflects their relatively high abundance in nature in environments frequented by humans, or a predilection for particularly thorny plant species. However, it also remains possible that certain species possess particular virulence factors that permit their immune evasion, prolonged survival/latency post-inoculation and the capacity to re-activate many years later if host immunity wanes due to immunosuppression or advancing years. 

## Figures and Tables

**Table 1 jof-08-00343-t001:** The causative agents and clinical presentations associated with the 106 cases of subcutaneous fungal infection/traumatic keratitis retrieved in database searches.

Organism (Number of Cases)	Clinical Presentation
* **Subcutaneous infections:** *	
*Alternaria* spp. (37)	Subcutaneous masses, animals (26)
	Subcutaneous nodules in humans (11) following SOT (6)
*Coccidioides immitis* (1)	Joint infection (knee)
*Curvularia* sp. (1)	Soft tissue infection
*Emarellia grisea* (1)	Fungal lesion, hand
*Exophiala* sp. (1)	Shin nodule
*Exophiala campbellii* (1)	Subcutaneous cyst
*Exophiala dermatitidis* (1)	Subcutaneous swelling and skin plaques, foot
*Exophiala jeanselmii* (1)	Subcutaneous mass
*Exophiala lecanii-corni* (1)	Cutaneous plaques and papules, arm
*Exophiala oligosperma* (1)	Subcutaneous nodules
*Exophiala xenobiotica* (3)	Infected ganglion (1), hand lesion (1), fungal mass in cat (1)
*Exserohilum rostratum* (1)	Nasal mass
*Falciformispora tompkinsii* (1)	Phaeohyphomycosis
*Fonsecaea monomorpha* (2)	Subcutaneous lesion (1), chromoblastomycosis (1)
*Fusarium oxysporum* complex (1)	Joint infection (knee)
*Fusarium solani* complex (1)	Eumycetoma
*Kirschsteiniothelia rostrata* (1)	Fungal abscess, foot.
*Leptospora* sp. (1)	Joint infection (knee)
*Madurella mycetomatis* (3)	Eumycetoma (3)
*Madurella pseudomycetomatis* (2)	Eumycetoma (2) incl. 1 in renal transplant patient
*Medicopsis romeroi* (6)	Eumycetoma (3)
	Foot “ulcer” (1)
	Arm lesion, post renal transplant (1)
	Foot lesion, post renal transplant (1)
*Microascus murinus* (1)	Foot lesion
*Ochroconis tshawytschae* (1)	Fungal keratitis
*Parathyridaria percutanea* (2)	Skin nodules, renal transplant patient (1), skin lesions elbows and knees (1)
*Phaeoacremonium* spp. (1)	Cystic lesions
*Phaeoacremonium parasiticum* (1)	Foot lesion
*Phaeoacremonium rubrigenum* (2)	Subcutaneous lesions in renal transplant patient (2)
*Phaeoisaria clematidis* (1)	Arm abscess
*Phialemoniopsis curvata* (1)	Skin nodule (foot), post renal transplant
*Plectosphaerella cucumerina* (1)	Tissue post tissue expanders
*Pseudallescheria ellipsoidea* (1)	Achilles abscess
*Rhytidhysteron rufulum* (1)	Ankle lesion
*Sarocladium bactrocephalum* (1)	Joint infection
*Scedosporium apiospermum* (10)	Subcutaneous nodules/lesions (7) post renal transplant (1)
	Joint infection (3)
*Scedosporium aurantiacum* (1)	Soft tissue infection
*Scedosporium de hoogii* (1)	Joint infection (1)
*Sporothrix brasiliensis* (1)	Skin lesion/ulcer after cat scratch
* **Traumatic keratomycosis:** *	
*Alternaria* spp. (5)	
*Cladophialophora boppii* (1)	
*Curvularia hawaiiensis* (1)	
*Scedosporium apiospermum* (3)	
*Scedosporium de hoogii* (1)	

**Table 2 jof-08-00343-t002:** Clinical presentation and mode of diagnosis of the 64 cases of infection caused by non-*Alternaria* species.

Organism (Case Number)	Identification Method	Site	Clinical Presentation
*Cladophialophora boppii*	rDNA sequencing of isolate	Corneal scrape	F 70y, query fungal keratitis
*Coccidioides immitis*	Phenotypic ID of isolate	Joint fluid	M 86y, Joint infection (knee)
*Curvularia* sp.	Phenotypic ID of isolate	Corneal scrape	M 40y, microbial keratitis
*Curvularia hawaiiensis*	MALDI-ToF MS of isolate	Foot tissue	M 68y, open dislocation of foot, RTA accident, Cambodia
* **Emarellia grisea** *	MALDI-ToF MS of isolate	Hand tissue	M 45y, Fungal lesion hand, site of previous eumycetoma
*Exophiala* sp.	Panfungal PCR of tissue	Elbow tissue	M 73y, keratotic, ulcerated lesion 6 months
* **Exophiala campbellii** *	MALDI-ToF MS of isolate	Subcutaneous cyst	F 68y, encapsulated palmar cyst, confirmed by PCR of tissue
* **Exophiala dermatitidis** *	MALDI-ToF MS of isolate	Tissue biopsy	M 58y, Subcutaneous swelling, confirmed by PCR of tissue
*Exophiala jeanselmii*	MALDI-ToF MS of isolate	Subcutaneous mass	F 45y, fluid aspirated from foot swelling
* **Exophiala lecanii-corni** *	MALDI-ToF MS of isolate	Skin biopsy	M 78y, cutaneous plaques/papules, arm; fungal elements seen
* **Exophiala oligosperma** *	MALDI-ToF MS of isolate	Skin biopsy	F 52y, subcutaneous nodule, shin; fungal elements seen x2, isolated twice 5 mo
***Exophiala xenobiotica*** (1)	MALDI-ToF MS of isolate	Tissue biopsy	F 11y CAT, right hind mass; fungal elements seen
*Exophiala xenobiotica* (2)	MALDI-ToF MS of isolate	Fluid, Hand	M 88y, infected ganglion
*Exophiala xenobiotica* (3)	Panfungal PCR of tissue	Tissue biopsy	M 69, large mass on arm
*Exserohilum rostratum*	Phenotypic ID of isolate	Nasal tissue	M 50y, right nasal fungal mass
* **Falciformispora tompkinsii** *	rDNA sequencing of isolate	Tissue biopsy	M 62y, phaeohyphomycosis of leg; confirmed by PCR of tissue, serum BDG +ve
*Fonsecaea monophora* (1)	rDNA sequencing of isolate	Tissue biopsy	M 74y, wrist lesion
***Fonsecaea monophora*** (2)	rDNA sequencing of isolate	Skin biopsy	M 59y, chromoblastomycosis, lesions > 20 y; histology positive
***Fusarium oxysporum* ***	Panfungal PCR of fluid	Knee fluid	F 25y, joint infection/ haematoma (knee); PCR positive for same organism 2 months later
***Fusarium solani* ***	Phenotypic ID of isolate	Tissue biopsy	M 47y, eumycetoma, foreign body removal; isolated twice
* **Kirschsteiniothelia rostrata** *	MALDI-ToF MS of isolate	Tissue biopsy	M 82y, fungal abscess, foot.; isolated 4 times over 4 months, fungal elements seen on each occasion
*Leptospora* sp.	Panfungal PCR of fluid	Synovial fluid	F 56y, joint infection, swollen knee
***Madurella mycetomatis*** (1)	Panfungal PCR of tissue	Finger biopsy	F 28y, black grain eumycetoma; grains seen in tissue, PCR +ve on second sample
***Madurella mycetomatis*** (2)	Panfungal PCR of tissue	Foot tissue	M 30y, chronic eumycetoma of foot; black grains seen
***Madurella mycetomatis*** (3)	MALDI-ToF MS of isolate	Foot biopsy	M 50y, eumycetoma of foot, previous penetrating injury, Sudan
***M. pseudomycetomatis*** (1)	rDNA sequencing of isolate	Foot tissue	F 67y, eumycetoma of foot 18 months, previous renal transplant
***M. pseudomycetomatis*** (2)	MALDI-ToF MS of isolate	Foot tissue	U 27y, eumycetoma surgically resected
***Medicopsis romeroi* (1)**	MALDI-ToF MS of isolate	Bone, foot	F 69y, eumycetoma previous diagnosis, treatment failure
***Medicopsis romeroi* (2)**	MALDI-ToF MS of isolate	Foot Tissue	M 60y, dark grain eumycetoma; isolated twice over 3 months
*Medicopsis romeroi* (3)	MALDI-ToF MS of isolate	Skin biopsy	M 68y, immunocompromised with foot “ulcer”
*Medicopsis romeroi* (4)	MALDI-ToF MS of isolate	Arm tissue	M 35y, arm lesion, post renal transplant
*Medicopsis romeroi* (5)	Panfungal PCR of tissue	Toe tissue	M 59y, soft tissue infection, post renal transplant
*Medicopsis romeroi* (6)	Panfungal PCR of tissue	Tissue/fluid	F 62y, no additional details given
* **Microascus murinus** *	rDNA sequencing of isolate	Biopsy, sole of foot.	M 58y, foot lesion; isolated twice
*Ochroconis tshawytschae*	rDNA sequencing of isolate	Eye	M 57y, Corneal infection
***Parathyridaria percutanea*** (1)	Panfungal PCR of tissue	Tissue biopsy	F 31y, skin nodules, renal transplant patient; pigmented fungal elements seen on histology
***Parathyridaria percutanea*** (2)	Panfungal PCR of tissue	Tissue, leg	F 50y, nodules knees and feet for 4 months; ex-Somalia, BDG +ve,
			Previous liver transplant, confirmed by isolation of organism
***Phaeoacremonium* spp.**	Phenotypic ID of isolate	Fluid, Toe and knee	M 63y, cystic lesions on toes, knee infection; isolated x5 over 6 month period
*Ph. parasiticum*	MALDI-ToF MS of isolate	Tissue foot	M 52y, mass forefoot > 2 yrs
*Ph. rubrigenum* (1)	MALDI-ToF MS of isolate	Aspirate, finger	F 71y, lesion index finger, renal transplant; isolated twice over 6 month period
***Ph. rubrigenum*** (2)	MALDI-ToF MS of isolate	Tissue biopsy	M 57y, thigh lesion, renal transplant; isolated x3 over 5 months
*Phaeoisaria clematidis*	rDNA sequencing of isolate	Tissue biopsy	M 46y, Abscess in arm; Nigerian
* **Phialemoniopsis curvata** *	MALDI-ToF MS of isolate	Tissue, foot	M 68y, nodule on foot, post renal transplant; confirmed by PCR of tissue, histology positive, Travel to Nigeria
*Plectosphaerella cucumerina*	MALDI-ToF MS of isolate	Tissue biopsy	M 32y, unwell post tissue expanders
*Pseudallescheria ellipsoidea*	MALDI-ToF MS of isolate	Tissue biopsy	M 41y, chronic Achilles abscess
* **Rhytidhysteron rufulum** *	Panfungal PCR of tissue	Tissue biopsy	M 33y, lump right ankle, suspect mycetoma; isolate also recovered
*Sarocladium bactrocephalum*	rDNA sequencing of isolate	Tissue biopsy	M 54y, joint infection
*Sced. apiospermum* (1)	MALDI-ToF MS of isolate	Fluid, finger	F 89y, index finger flexor sheath abscess
*Sced. apiospermum* (2)	MALDI-ToF MS of isolate	Fluid, joint	F 74y, septic arthritis
*Sced. apiospermum* (3)	MALDI-ToF MS of isolate	Fluid, knee	F 75y, intrapatellar bursitis
*Sced. apiospermum* (4)	MALDI-ToF MS of isolate	Tissue biopsy	F 68y, cellulitis forearm, post renal transplant
*Sced. apiospermum* (5)	MALDI-ToF MS of isolate	Tissue swab	M 70y, enlarging sore after gardening wound
*Sced. apiospermum* (6)	MALDI-ToF MS of isolate	Tissue, forearm	M 68y, multiple abscesses forearm, glioblastoma
*Sced. apiospermum* (7)	MALDI-ToF MS of isolate	Tissue, leg	M 83y, cellulitis right leg
***Sced. apiospermum*** (8)	MALDI-ToF MS of isolate	Tissue, hand	M 76y, Right hand wound, AML; isolated twice over 3 weeks
*Sced. apiospermum* (9)	MALDI-ToF MS of isolate	Tissue, thigh	F 71y, thigh lesion
*Sced. apiospermum* (10)	MALDI-ToF MS of isolate	Tissue, wrist	F 60y, dorsal wrist mass and synovitis > 2 yrs
*Sced. apiospermum* (11)	MALDI-ToF MS of isolate	Corneal scrape	F 40y, fungal keratitis
*Sced. apiospermum* (12)	MALDI-ToF MS of isolate	Corneal scrape	F 65y, superficial fungal keratitis
*Sced. apiospermum* (13)	MALDI-ToF MS of isolate	Eye	M 63y, no additional clinical details provided
* **Scedosporium aurantiacum** *	MALDI-ToF MS of isolate	Tissue, ankle	F 63y, fungal mass, wound rotten wood; isolated twice over 7 d
*Scedosporium de hoogii* (1)	MALDI-ToF MS of isolate	Aspirate, wrist	M 81y, infected wrist joint, “low immunity”
*Scedosporium de hoogii* (2)	MALDI-ToF MS of isolate	Corneal scrape	F 67y, fungal keratitis with corneal ulcer
*Sporothrix brasiliensis*	rDNA sequencing of isolate	Pus	M 28y, skin lesions/ulcers after cat scratch; Brazilian cat imported into UK, histology positive

Bold isolates are proven infections; * denotes identification to species complex level. BDG = serum β-D-Glucan test. For isolates identified by rDNA sequencing, generated sequences matched the following GenBank accession numbers with the given % identity: *Cladophialophora boppii* (HQ114280; 100%); *Falciformispora tompkinsii* (NR_132041; >98.5%); *Fonsecaea monophora* (LC317599; >98.5%); *Madurella pseudomycetomatis* (MK926823; >99%); *Microascus murinus* (MH871664; 99%); *Ochroconis tshawytschae* (MH870422; 99%); *Phaeoisaria clematidis* (MW131990; >97.5%); *Sarocladium bactrocephalum* (MH859409; 98.5%); *Sporothrix brasiliensis* (MH877527; 100%).

**Table 3 jof-08-00343-t003:** MIC values for 50 non-*Alternaria* species.

Organism	MIC (mg/L)						
	AMB	VRC	ITC	PSC	AND	NAT	OTHER
*Cladophialophora boppii*	1	0.5	nd	nd	nd	4	-
*Curvularia* sp.	0.25	1	0.25	nd	nd	1	-
*Curvularia hawaiiensis*	0.125	0.5	nd	0.06	nd	nd	-
*Emarellia grisea*	0.5	0.125	0.25	0.25	nd	nd	-
*Exophiala campbellii*	0.5	0.125	0.25	0.125	<0.015	nd	-
*Exophiala jeanselmei*	1	0.25	0.25	nd	nd	nd	-
*Exophiala lecanii-corni*	1	0.25	0.5	0.25	nd	nd	ISC = 2
*Exophiala oligosperma*		0.5	0.5	0.06	0.06	nd	nd
*Exophiala xenobiotica* (1)	0.5	4	0.25	nd	nd	nd	-
*Exophiala xenobiotica* (2)	0.25	0.25	0.25	nd	nd	nd	-
*Exserohilum rostratum*	0.25	1	0.5	nd	0.06	nd	-
*Fonsecaea monomorpha*	0.5	0.06	0.25	0.06	nd	nd	ISC = 0.06
*Fusarium solani*	4	4	>16	nd	nd	nd	-
*Kirschsteiniothelia rostrata*	1	0.06	0.25	0.06	<0.015	nd	ISC = 0.25
*Madurella mycetomatis*	0.125	0.25	0.125	nd	4	nd	-
*Madurella pseudomycetomatis*	0.125	0.125	0.25	0.06	4	nd	-
*Medicopsis romeroi* (1)		1	0.25	1	nd	0.5	nd
*Medicopsis romeroi* (2)	0.5	2	1	1	2	nd	TRB = 0.125
*Medicopsis romeroi* (3)	0.5	0.125	0.06	0.06	nd	nd	ISC = 0.25
*Medicopsis romeroi* (4)	nd	1	>16	0.25	nd	nd	TRB = 0.06
*Microascus murinus*	0.5	8	0.5	nd	0.5	nd	-
*Parathyridaria percutanea*	0.25	2	>16	nd	2	nd	-
*Phaeoacremonium* spp.	0.5	0.125	4	nd	8	nd	-
*Phaeoacremonium rubrigenum* (1)	0.5	0.125	1	nd	nd	2	-
*Phaeoacremonium rubrigenum* (2)	1	0.25	0.5	0.25	nd	nd	ISC = 1
*Phaeoacremonium parasiticum*	1	0.5	>16	0.5	8	nd	ISC = 0.5
*Phaeoisaria clematidis*	0.5	0.06	0.25	0.06	0.125	nd	TRB = 0.125
*Phialemoniopsis curvata*	nd	0.5	1	0.5	nd	nd	-
*Plectosphaerella cucumerina*	0.5	2	0.5	nd	nd	nd	-
*Pseudallescheria ellipsoidea*	nd	nd	0.5	nd	nd	nd	NYT = 4
*Rhytidhysteron rufulum*	0.5	0.25	0.06	<0.03	nd	nd	-
*Sarocladium bactrocephalum*	0.125	0.125	0.25	nd	4	nd	-
*Scedosporium apiospermum* (1)	1	0.5	4	nd	2	nd	-
*Scedosporium apiospermum* (2)	4	0.5	1	nd	nd	nd	-
*Scedosporium apiospermum* (3)	4	0.25	0.25	nd	nd	nd	-
*Scedosporium apiospermum* (4)	2	0.5	1	nd	nd	nd	TRB = >16
*Scedosporium apiospermum* (5)	4	0.5	0.5	0.5	nd	nd	-
*Scedosporium apiospermum* (6)	4	1	2	2	nd	nd	-
*Scedosporium apiospermum* (7)	0.5	0.125	0.5	nd	nd	nd	-
*Scedosporium apiospermum* (8)	2	0.5	2	0.5	nd	nd	-
*Scedosporium apiospermum* (9)	4	0.125	0.25	nd	nd	nd	-
*Scedosporium apiospermum* (10)	1	0.5	1	nd	nd	nd	-
*Scedosporium apiospermum* (11)	2	0.5	nd	nd	nd	2	-
*Scedosporium apiospermum* (12)	4	0.5	4	nd	nd	4	ECZ = 2
*Scedosporium apiospermum* (13)	2	0.5	0.5	nd	nd	2	ECZ = 0.5
*Scedosporium aurantiacum*	16	0.5	2	0.5	nd	nd	-
*Scedosporium de hoogii* (1)	8	0.25	1	1	8	nd	-
*Scedosporium de hoogii* (2)	2	0.25	0.5	nd	nd	2	-
*Sporothrix brasiliensis*	0.5	16	0.5	0.5	nd	nd	ISC = 4

Abbreviations: nd = not done; AMB = amphotericin B; VRC = voriconazole; ITC = itraconazole; PSC = posaconazole; AND = anidulafungin; NAT = natamycin; TRB = terbinafine; ISC = isavuconazole; ECZ = econazole; NYT = nystatin.

**Table 4 jof-08-00343-t004:** Clinical details and diagnosis method for the 42 cases of subcutaneous infection and keratitis caused by *Alternaria* spp.

Case (Year)	ID	Site	Clinical Presentation
**Isolates recovered from clinical material and susceptibility testing was requested/performed:**			
1 (2016)	PT	Tissue, knee	M 66y, right knee lesion following renal transplant
2 (2017)	rD	Tissue, thigh	M 57y, lesion right thigh following hand transplant
3 (2017)	MT	Tissue	M 66y, subcutaneous phaeohyphomycotic cyst
4 (2020)	MT	Tissue, thigh	M 69y, purple nodule on right thigh
5 (2019)	MT	Tissue, knee	F 36y, left knee ulcer following renal transplant
6 (2018)	PT	Corneal scrape	F 23y, fungal keratitis
7 (2020)	PT	Corneal scrape	M 52y, fungal keratitis
8 (2018)	PT	Corneal scrape	F 66y, fungal keratitis
9 (2020)	MT	Tissue, nose	F, CAT, dermatitis on nose, fungal elements seen
10 (2017)	PT	Tissue, ear	F, CAT, fungal granuloma excised from ear pinna, fungal elements seen
11 (2020)	PT	Tissue	M, CAT, recurrent pyogranulomatous dermatitis with abundant fungal elements
			Confirmed by PCR on tissue
12 (2020)	PT	Skin biopsy	F, DOG, multiple skin nodules
**Isolates recovered from clinical material but susceptibility testing was not requested:**			
13 (2021)	rD	Tissue	F 67y, no clinical details provided
14 (2019)	PT	Corneal scrape	M 59y, corneal graft following chemical injury
15 (2021)	MT	Corneal scrape	M 59y, keratitis following corneal abrasion
16 (2019)	PT	Tissue biopsy	U, CAT, Fungal mycetoma
17 (2017)	PT	Tissue biopsy	M, CAT, cutaneous fungal granuloma on nose
18 (2021)	PT	Tissue biopsy	M 4y, CAT, swollen nose, fungal elements seen
19 (2021)	PT	Tissue biopsy	F 6y, CAT. Ulcerated nodule nose, fungal elements seen
20 (2016)	PT	Tissue biopsy	M CAT, nasal mass, fungal elements seen
21 (2017)	PT	Tissue, nose	M, CAT, fungal granuloma on nose, fungal elements seen
			Independent isolation from second biopsy
22 (2018)	PT	Tissue, nose	M, CAT, nasal mass, fungal elements seen
23 (2020)	PT	Tissue, nose	U, CAT, fungal granuloma on nose
24 (2017)	PT/rD	Tissue, leg	F, DOG, immunosuppressed, multifocal abscesses on legs and paws,
			Phaeohyphomycosis on histology. Confirmed by direct PCR on tissue, two furtherindependent isolations, direct microscopy positive on all samples
25 (2017)	rD	Tissue, nose	M, CAT, fungal mass bridge of nose, fungal elements seen on histology and direct
			examination of tissue
26 (2019)	PT	Tissue, foot	M, CAT, metatarsal and digital lesions, fungal elements seen on direct examination
27 (2017)	PT	Tissue	M, CAT, fungal mass, mycotic granuloma on histology, fungal elements seen on
			direct examination
**Diagnosis made solely by panfungal PCR of tissue (no isolate was recovered for further analyses):**			
28 (2020)	PCR	Tissue, thigh	M 59y, thigh lesion following renal transplant
29 (2019)	PCR	Skin Biopsy	U, 57y, no clinical details provided
30 (2016)	PCR	Tissue, leg	M 39y, leg lesion following renal transplant
31 (2018)	PCR	Tissue Biopsy	F 61y, granulomatous fungal folliculitis post-transplant
32 (2017)	PCR	Skin biopsy	M 64y, Crohns, purple ulcerated papules following rose thorn injury
33 (2016)	PCR	Tissue	M, CAT, no clinical details provided
34 (2017)	PCR	Tissue, nose	M, CAT, no clinical details provided
35 (2018)	PCR	Tissue, foot	M, CAT, solitary cutaneous mass on paw, histology-confirmed fungal granuloma
36 (2020)	PCR	Tissue, nose	M, CAT, nasal mass >2yrs, recurred post-antifungal treatment (itraconazole)
37 (2020)	PCR	Punch biopsy	M, CAT, no clinical details provided. Repeat PCR positive on separate sample
38 (2020)	PCR	Biopsy, ear	M, DOG, mass left pinna
39 (2020)	PCR	Biopsy	M 5y, CAT. no clinical details provided
40 (2020)	PCR	Biopsy, nose	M 11y, CAT, cutaneous fungal granuloma on muzzle
41 (2021)	PCR	Biopsy, nose	M 11y, DOG, nose mass, mycotic granuloma associated with foreign material
42 (2021)	PCR	Biopsy, nose	M 9y, CAT, nasal mass

ID = identification/diagnosis method; MT = MALDI-ToF MS of isolates; PT = phenotypic identification of isolate; rD = rDNA sequencing of isolate; PCR = panfungal PCR of tissue. For isolates identified by rDNA sequencing, individual sequences matched sequences corresponding to a variety of “*Alternaria* sp.” in GenBank, always with >98.5% sequence identity.

**Table 5 jof-08-00343-t005:** Antifungal MIC profiles for the *Alternaria* isolates 1–12 from Table 4. Antifungal drug abbreviations are as described in Table 3.

Isolate Number	MIC (mg/L)							
	AMB	ITC	VRC	PSC	ISC	AND	TRB	NAT
1	0.25	0.5	4	0.06	4	<0.015	0.25	2
2	0.25	0.5	8	0.06	16	<0.015	4	nd
3	0.5	0.5	<0.03	0.25	nd	nd	4	nd
4	0.5	0.25	2	0.125	nd	0.125	nd	nd
5	0.5	0.5	16	0.5	16	nd	0.5	nd
6	0.5	0.25	0.5	nd	nd	nd	nd	2
7	0.25	0.5	2	0.25	nd	nd	nd	2
8	1	1	8	0.25	nd	nd	nd	2
9	0.5	0.5	8	nd	nd	nd	nd	nd
10	0.5	0.25	8	nd	nd	nd	nd	nd
11	nd	0.25	4	nd	nd	nd	nd	nd
12	nd	0.5	nd	nd	nd	nd	0.125	nd

## Data Availability

No data were reporetd in this study.

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
