# Peer review of "Fungal Infections of Implantation: More Than Five Years of Cases of Subcutaneous Fungal Infections Seen at the UK Mycology Reference Laboratory"

_jof, 2022, doi:10.3390/jof8040343_

Round 1

Reviewer 1 Report

This manuscript presents a compilation of uncommon cases of fungal cutaneous infections post-inoculation. This is an extensive well-written description of rare cases, which occurred over several years. The added value of this study is that it provides interesting data on antifungal susceptibility to various antifungals within a large panel of fungal isolates. Noteworthy, these fungi often growing slowly, thus MICs are not easy to determine in mycology routine labs.

I raised a few missing points, which should be addressed:

Materials and methods:

Data about molecular diagnosis and especially panfungal PCR are lacking. The reference 24 (Simpson, Vet Rec 2009) which is cited seems to be wrong, as there is no mention of PCR. Please precise which primers and which region was targeted by panfungal PCR, as it could impact identification accuracy.

Results

  • Tables 2 and 4: Regarding cases diagnosed in SOT patients, I would suggest to expand a bit more the description of clinical background: Could you precise the delay between transplantation and the onset of infection? Could you mention the type of immunosuppressive therapy? Indeed, the intensity of immunosuppression can influence the reactivation and/or enlargement of a latent lesion at the inoculation point.
  • Moreover, could you provide details on the context of inoculation? Specific outdoor activities? gardening? other…??
  • In addition, I think the presumed geographical area of contamination is missing, for example in Table 1 (at least the continent?)
  • Table 2: according to the description of proven cases (lines 217-221), a sole positive panfungal PCR (even positive twice) is not enough to classify as proven case, as it needs to be associated with positive culture or histological evidence. Please, check the Fusarium oxysporum case which has been classified as proven…
  • Table 3: In the title, correct the number of tested isolates, as there are less than 64 tested isolates: only 50?
  • Please precise if all MICs were performed on isolates from patients with no previous exposure to antifungals. Were some strains isolated and/or sent to your lab because of treatment failure? (This important data is only specified for SOT-isolated strains)
  • Line 297: regarding susceptibility to anidulafungin, please precise in the text that only 3/12 Alternaria isolates have been tested
  • Line 264, clarify the clinical origin of the 42 Alternaria cases: 26 in domestic animals, 10 subcutaneous nodules in humans and 5 fungal keratitis (totalizing 41) => one case is missing (and it is discordant with line 319 “11 cases”)

Minor corrections:

Table 2: Modify “PCR positive 2 for same organism months later” on Fusarium oxysporum’ line.

Line 245: “when” instead of “where”?

Line 268: move the coma after “keratitis”

Table 4 footnote: Please correct “Tabler 4” “panfuingal”

Line 298: “posaconzole”

Author Response

Data about molecular diagnosis and especially panfungal PCR are lacking. The reference 24 (Simpson, Vet Rec 2009) which is cited seems to be wrong, as there is no mention of PCR. Please precise which primers and which region was targeted by panfungal PCR, as it could impact identification accuracy.

This has now been corrected, with additional details and references provided. Many apologies.

Results

  • Tables 2 and 4: Regarding cases diagnosed in SOT patients, I would suggest to expand a bit more the description of clinical background: Could you precise the delay between transplantation and the onset of infection? Could you mention the type of immunosuppressive therapy? Indeed, the intensity of immunosuppression can influence the reactivation and/or enlargement of a latent lesion at the inoculation point.
  • Moreover, could you provide details on the context of inoculation? Specific outdoor activities? gardening? other…??
  • In addition, I think the presumed geographical area of contamination is missing, for example in Table 1 (at least the continent?)
  • All of the above points rae now covered in a new "limitations" section in the discussion as follows: "There are several limitations to the current study. It is unfortunate that for many of the cases described here, detailed history of previous trauma, occupation of the patients and geographical area of likely acquisition are lacking. In addition, for many of the solid organ transplantation patients, data concerning delay between transplantation and onset of clinical presentation and the nature and duration of immunosuppressive agents employed was lacking. In part, this reflects the nature of national reference laboratory work, which is based on patient/case referrals with limited direct access to patient data or the ability to seek additional clinical information. However, it is also a likely reflection of the fact that many subcutaneous infections present clinically many years after initial acquisition, following relatively minor or innocuous traumas that the patient does not recollect. This situation is probably aggravated in those cases that follow later solid organ transplantation, where inoculated organism has remained inert/subclinical for many years.  It is also unfortunate that for most of the cases, we cannot be certain that organisms submitted to antifungal susceptibility testing were from patients that were antifungal treatment naïve. However, given that these isolates were either referred to our laboratory from diagnostic biopsy specimens, or cultured from them here at the MRL, it is likely that the majority of these cases represent the initial presentation/diagnosis and theta the patients had not received prior antifungal treatment (with the exception of the relapsed cases of E. grisea and M. romeroi (case 1) infection (Table 2). "
  • Table 2: according to the description of proven cases (lines 217-221), a sole positive panfungal PCR (even positive twice) is not enough to classify as proven case, as it needs to be associated with positive culture or histological evidence. Please, check the Fusarium oxysporum case which has been classified as proven…
  • Many apologies, this has now been amended.
  • Table 3: In the title, correct the number of tested isolates, as there are less than 64 tested isolates: only 50? Sorry once again, now corrected to state 50.
  • Please precise if all MICs were performed on isolates from patients with no previous exposure to antifungals. Were some strains isolated and/or sent to your lab because of treatment failure? (This important data is only specified for SOT-isolated strains). We think the patients were mostly antifungal naive; this points is discussed in the new "limitations" section.
  • Line 297: regarding susceptibility to anidulafungin, please precise in the text that only 3/12 Alternaria isolates have been tested. This information has now been added.
  • Line 264, clarify the clinical origin of the 42 Alternaria cases: 26 in domestic animals, 10 subcutaneous nodules in humans and 5 fungal keratitis (totalizing 41) => one case is missing (and it is discordant with line 319 “11 cases”). Many apologies; the correct number of cases is now stated: 42 including 11 subcutaneous infections in humans 

Minor corrections:

Table 2: Modify “PCR positive 2 for same organism months later” on Fusarium oxysporum’ line. Corrected as requested.

Line 245: “when” instead of “where”? Corrected as requested.

Line 268: move the coma after “keratitis” Corrected as requested.

Table 4 footnote: Please correct “Tabler 4” “panfuingal” Corrected as requested.

Line 298: “posaconzole” Corrected as requested.

Reviewer 2 Report

I read the manuscript with interest and congratulate the authors for writing a manuscript taking into consideration ‘One Health Approach’ looking for infections both in humans and animals.

Some suggestions:

Line 102-103: Association with renal transplantation is an observation and not objective of study and so, may not be highlighted here

Line 108: time period is not 5 years exactly

Line 120 to throughout manuscript:

No doubt keratitis is most commonly causes by trauma leading to implantation of fungal element or so…..but were these all cases of keratomycosis ??

If yes, even then, keratomycosis is an entirely different entity, identified more easilyand treated more efficiently than other subcutaneous infections. To understand other subcutaneous presentations and dilemma and delay  in their diagnosis, I suggest authors should make two groups: keratomycosis and other subcutaneous  presentations , so that the etiological agents and their management could be understood in a better way.

Even mycetoma cases which typically present with triad of symptoms (tumefaction, draining sinuses and grains) can be separated out. Typically different etiological agents……

Similarly, animal and human infection groups should be separate for better understanding of readers.

Line 145 and 157: cases were 106 or 105 ?

Line 294: any reason for low culture positivity of alternaria ? It grows easily in culture, rather a common laboratory contaminant. If identified by molecular methods, how did the authors differentiated in possibility of contamination and true pathogen ?

Any history of trauma/ type of occupation for these patients , if available can be analyzed (farm background, pet animals, geographical area of animals affected  etc)   

Line 399: 6 years of data  ?  mismatch with title

Thanks

Author Response

I read the manuscript with interest and congratulate the authors for writing a manuscript taking into consideration ‘One Health Approach’ looking for infections both in humans and animals.

Some suggestions:

Line 102-103: Association with renal transplantation is an observation and not objective of study and so, may not be highlighted here

This sentence has been deleted from the revised manuscript.

Line 108: time period is not 5 years exactly

This has been homogenised throughout the entire manuscript (including the title) to read >5 years.

Line 120 to throughout manuscript:

No doubt keratitis is most commonly causes by trauma leading to implantation of fungal element or so…..but were these all cases of keratomycosis ??

If yes, even then, keratomycosis is an entirely different entity, identified more easilyand treated more efficiently than other subcutaneous infections. To understand other subcutaneous presentations and dilemma and delay  in their diagnosis, I suggest authors should make two groups: keratomycosis and other subcutaneous  presentations , so that the etiological agents and their management could be understood in a better way.

We believe that all of these cases of keratitis were traumatic inoculation. However, we agree with the reviewer and have now revised Table 1 to separate cases/organisms according to subcutaneous versus traumatic ketatomycosis.

Even mycetoma cases which typically present with triad of symptoms (tumefaction, draining sinuses and grains) can be separated out. Typically different etiological agents…… Here we disagree. Several of the cases of subcutaneous infection caused by agents of eumycetom (Scedosporium, Medicopsis romeroi) did not present with the typical features of eumycetoma; we have seen this picture frequently, especially in cases of SOT, and this is discussed in the revised manuscript, In addition, one of the other reviewers has suggested combining several of the existing Tables rather than separating them further, so it is impossible to accommodate both of these conflicting reviewer suggestions.

Similarly, animal and human infection groups should be separate for better understanding of readers. One of the other reviewers has suggested combining several of the existing Tables rather than separating them further, so it is impossible to accommodate both of these conflicting reviewer suggestions.

Line 145 and 157: cases were 106 or 105 ? Many apologies; this has now been corrected throughout.

Line 294: any reason for low culture positivity of alternaria ? It grows easily in culture, rather a common laboratory contaminant. If identified by molecular methods, how did the authors differentiated in possibility of contamination and true pathogen ?

This is now discussed in a new paragraph in the revised manuscript as follows: "The apparent relatively low rate of recovery of Alternaria spp. isolates (12 isolates from 42 cases) is a reflection of the sample types that were received in many cases, where clinical material was submitted after previous formalin fixation, or as wax curls from histological blocks, thus preventing attempts at culture. In most cases, examination of material by direct microscopy at the MRL, or histology reports that accompanied the samples revealed or reported fungal elements consistent with the final diagnosis, making it unlikely that these molecular diagnoses were detecting fungal contaminants rather than the true pathogen (data not shown)."

Any history of trauma/ type of occupation for these patients , if available can be analyzed (farm background, pet animals, geographical area of animals affected  etc)   This has been addressed in a new paragraph, along with suggestions from reviewer 1, entitled limitations as follows: "There are several limitations to the current study. It is unfortunate that for many of the cases described here, detailed history of previous trauma, occupation of the patients and geographical area of likely acquisition are lacking. In addition, for many of the solid organ transplantation patients, data concerning delay between transplantation and onset of clinical presentation and the nature and duration of immunosuppressive agents employed was lacking. In part, this reflects the nature of national reference laboratory work, which is based on patient/case referrals with limited direct access to patient data or the ability to seek additional clinical information. However, it is also a likely reflection of the fact that many subcutaneous infections present clinically many years after initial acquisition, following relatively minor or innocuous traumas that the patient does not recollect. This situation is probably aggravated in those cases that follow later solid organ transplantation, where inoculated organism has remained inert/subclinical for many years.  It is also unfortunate that for most of the cases, we cannot be certain that organisms submitted to antifungal susceptibility testing were from patients that were antifungal treatment naïve. However, given that these isolates were either referred to our laboratory from diagnostic biopsy specimens, or cultured from them here at the MRL, it is likely that the majority of these cases represent the initial presentation/diagnosis and theta the patients had not received prior antifungal treatment (with the exception of the relapsed cases of E. grisea and M. romeroi (case 1) infection (Table 2)."

Line 399: 6 years of data  ?  mismatch with title

This has been homogenised throughout to read >5 years of data

Reviewer 3 Report

  The article is very well written, easy to read and scientifically relevant and it is recommended for publication. The  only suggestion would be to merge the contents of tables 1 and 2, which seem redundant. Moreover, it is necessary to add to the data of the isolates  identified by rDNA sequencing  the deposit number of the sequences in the GenBank; and  or add an asterisk to this information presented in the table  and present this data in the table legend.

Author Response

The article is very well written, easy to read and scientifically relevant and it is recommended for publication.

Many thanks for your kind words.

The  only suggestion would be to merge the contents of tables 1 and 2, which seem redundant.

Unfortunately, reviewer 2 made the opposite requests: to separate the Tables more: to separate cases of  keratitis/ subcutaneous/eumycetoma (which we have partially addressed by separating out cases of keratitis) .  It is impossible to successfully accommodate such conflicting suggestions.

Moreover, it is necessary to add to the data of the isolates  identified by rDNA sequencing  the deposit number of the sequences in the GenBank; and  or add an asterisk to this information presented in the table  and present this data in the table legend.

This information is now included in the Table legends. It is not the practise of our laboratory to submit hundreds of DNA sequences to GenBank for infections/organisms that were only identified by DNA and not published as individual case reports with full supporting histories and phenotypic corroboration. This is one of the reasons historically (that other authors do submit such sequences)  that has led to fungal DNA sequences in GenBank being often of dubious quality/validity.  Instead, here we have chosen to provide in the Legends the EMBL accession numbers and percent sequence identities that our DNA sequences matched during BlastN searches.